# Uniformly Dispersed Nano Pd-Ni Oxide Supported on Polyporous CeO$_2$ and Its Application in Methane Conversion of Tail Gas from Dual-Fuel Engine

Chunlian Luo [1,2,3], Luwei Chen [4] ([ID]), Abdullah N. Alodhayb [5] ([ID]), Jianhua Wu [1,2], Mingwu Tan [4,*] and Yanling Yang [1,2,*]

1 College of Marine Engineering, Jimei University, Xiamen 361000, China; clianluo@163.com (C.L.); wujh@jmu.edu.cn (J.W.)
2 Key Laboratory for Marine Corrosion and Intelligent Protection Materials of Xiamen, Jimei University, Xiamen 361000, China
3 Navigation College, Xiamen Ocean Vocational College, Xiamen 361012, China
4 Institute of Sustainability for Chemicals, Energy and Environment (ISCE2), Agency for Science, Technology and Research (A*STAR), 1 Pesek Road, Jurong Island, Singapore 627833, Singapore; chen_luwei@isce2.a-star.edu.sg
5 Department of Physics and Astronomy, College of Science, King Saud University, Riyadh 11451, Saudi Arabia; aalodhayb@ksu.edu.sa
* Correspondence: tan_mingwu@isce2.a-star.edu.sg (M.T.); yangyl@jmu.edu.cn (Y.Y.)

**Abstract:** The development of catalysts for low-temperature methane combustion is crucial in addressing the greenhouse effect. An effective industrial catalyst strategy involves optimizing noble metal utilization and boosting metal–metal interaction. Here, the PdNi-H catalyst was synthesized using the self-assembly method, achieving the high dispersion and close proximity of Pd and Ni atoms compared to the counterparts prepared by the impregnation method, as confirmed by EDS mapping. The XRD and TEM results revealed Pd$^{2+}$ and Ni$^{2+}$ doping within the CeO$_2$ lattice, causing distortions and forming Pd-O-Ce or Ni-O-Ce structures. These structures promoted oxygen vacancy formation in CeO$_2$, and this was further confirmed by the Raman and XPS results. Consequently, the PdNi-H catalyst demonstrated an excellent redox ability and catalytic activity, achieving lower ignition and complete methane burning temperatures at 282 and 387 °C, respectively. The highly dispersed PdNi species played a pivotal role in activating methane for enhanced redox ability. Additionally, the narrow size distribution range contributed to more vacancies on the surface of CeO$_2$, as confirmed by the XPS results, thereby facilitating the activation of gas phase oxygen to form oxygen species (O$_2^-$). This collaborative catalytic approach presents a promising strategy for developing efficient and stable methane combustion catalysts at low temperatures.

**Keywords:** Pd-Ni alloy oxide; ceria; co-doping; oxygen vacancy; methane combustion





## 1. Introduction

Methane (CH$_4$), a primary component of natural gas and shale gas, possesses a high hydrogen-to-carbon ratio. This quality makes it an environmentally friendly alternative fuel for engine and heating devices, potentially relieving fossil fuel resource scarcities and reducing pollution associated with traditional diesel engine emissions. However, the release of unconverted methane into the environment accelerates the greenhouse effect, leading to an impact of more than 20 times higher than that of CO$_2$ [1]; thus, the conversion of residual unburned methane in exhaust gas is essential. Although methane offers a green fuel option, its stable chemistry needs high-temperature combustion, resulting in the generation of harmful gas such as NO$_x$ and SO$_2$. Therefore, the development of catalysts capable of completing the combustion of methane at low temperatures becomes imperative.

Numerous research efforts have already been put into this pursuit, resulting in substantial progress [2–4].

Pd-based catalysts have exhibited exceptional activity in methane combustion due to the lower activation energy of the C-H bond, with the Pd-PdO$_x$ phase acknowledged as a highly active site [5]. Typically, the synergy between the oxygen vacancy on the support surface and Pd species enhances the catalytic activity of the Pd catalysts [6]. For example, Hoffmann et al. [7] developed a simple porous glass integration method to synthesize a PdO/CeO$_2$ catalyst (0.63 wt% Pd loading), which exhibited good performances for the complete burning of methane at 350 °C. It was reported that the introduction of metal ions into CeO$_2$ greatly reduced the formation energy of oxygen vacancies [8,9]. The creation of the surface oxygen vacancy promotes the activation of oxygen [10–12], thereby driving methane combustion. In addition, the amount of effectively active sites is also the key to affecting the activity of the Pd catalyst. Chen et al. [13] reported that highly dispersed Pd on nanorod-like phosphorus-doped mesoporous γ-alumina exhibited excellent catalytic performances for methane combustion at a temperature as low as 345 °C; however, the agglomeration of active sites led to the compromised thermal stability during the reaction. In terms of this issue, Duan et al. [14] encapsulated and stabilized Pd/SiO$_2$ particles with porous Al$_2$O$_3$ overlayers through the atomic layer deposition (ALD) method, while the cumbersome operation of such a catalyst preparation resulted in an additional cost. Therefore, using a simple but effective method to achieve highly active and stable Pd-based catalysts to realize the combustion of methane at low temperatures is the key to achieving wide application.

Meanwhile, considering the cost of Pd, employing non-precious metals to form binary alloys is a potential strategy to enhance noble metals' utilization. Shen et al. [15] prepared PdNi-based catalysts, wherein Pd was modified with a minute amount of Ni, allowing the combustion of methane at 430 °C. Zou et al. [16] fabricated a catalyst, NiO@PdO/Al$_2$O$_3$, with a 0.2 wt.% Pd loading and a molar Ni/Pd ratio of 2/1, exhibiting 99% CH$_4$ conversion and sustained stability at 400 °C, possibly owing to the enhanced NiO-PdO interfacial interaction. However, it is noteworthy that a decrease in activity due to the addition of a secondary metal is also widely reported because the formation of a partially oxidized PdO phase was prevented. Furthermore, different manufacturing methods have a greater impact on the combination of the two metals, resulting in various moieties, such as isolated components, alloys, core–shell, mushroom-like, and so on [16–19]. Therefore, there is an urgent need to develop highly dispersed Pd sites where Ni species are tightly bound and to improve durability without sacrificing activity.

In this work, we used cerium dioxide as a carrier to support Pd-Ni species instead of pure Pd or Ni. A novel synthesis approach was developed to facilitate the dispersion of PdO and NiO and strengthen their interaction. Firstly, we used a pyrolysis method to prepare PdNi binary alloy nanoparticles in a uniform size with the protective agent of TPO (tri-n-octylphosphine), which led to a hydrophobic surface. Afterward, the PdNi alloy nanoparticles were dispersed on a fabricated polyporous CeO$_2$. After calcination, we obtained the PdNi-H sample (with a 0.5 wt% Pd loading and a molar Ni/Pd ratio of 5/1). It showed an excellent performance with the complete combustion of methane at 387 °C and stability in longtime test at 350 °C. Such a designing strategy of the catalysts by promoting bimetallic interaction is inspiring for improving noble metal utilization efficiency in CH$_4$ oxidation and other related reaction systems.

## 2. Results and Discussion

### 2.1. Physicochemical Property of Metal Nanoparticles and Supported Samples

In order to enhance the interaction between Pd and Ni species, the PdNi alloy was synthesized with a 1:5 Pd and Ni ratio and investigated to obtain more information on the advantage of the designed self-assembly method, as well as the pure metal nanoparticles. Figure 1a–c show TEM and high-resolution TEM (inset) images of pure Pd, Ni, and PdNi alloy nanoparticles with uniform size. According to the inserted HRTEM images, the lattice

spacing of pure Pd and Ni nanoparticles (NPs) is 0.24 nm and 0.21 nm, corresponding to the Pd (111) and Ni (111) lattice planes, and for the PdNi nanoparticles, it is 0.22 nm, which intuitively demonstrates the formation of bimetallic alloy according to the literature report [20]. The responding statistical results of the particle size distribution are shown in Figure 2d–f, and the average Pd, Ni, and PdNi particle sizes are 2.8, 7.2, and 6.9 nm, in which the bimetallic nanoparticle size is situated between pure Pd and Ni. Simultaneously, the XRD pattern of PdNi nanoparticles in Figure 2a shows that the diffraction peak at $2\theta = 42.3°$ was located between pure Ni (ICDD-JCPDS Card No. 04-0850) at $2\theta = 44.8°$ and pure Pd (ICDD-JCPDS Card No. 46-10430) at $2\theta = 40.0°$ [21], further indicating that the PdNi alloy is formed, in accordance with TEM result.

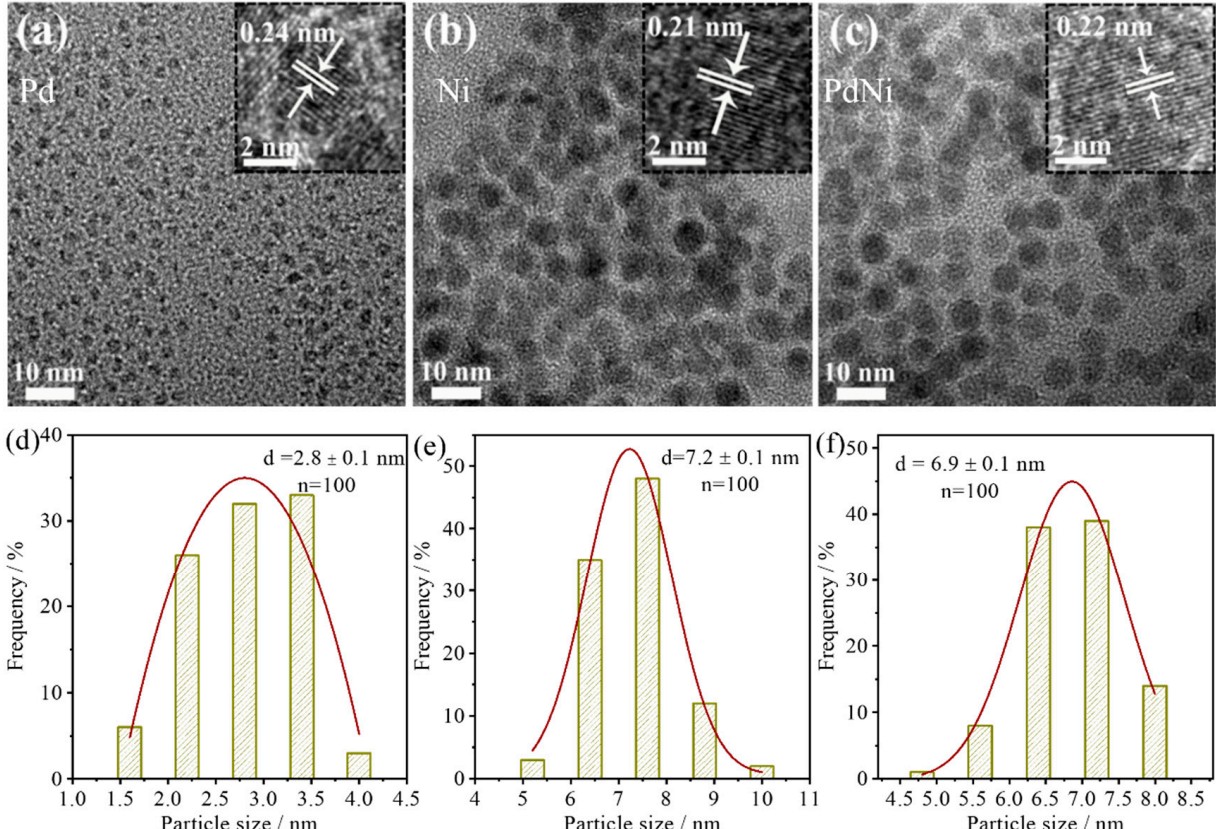

**Figure 1.** TEM and HRTEM images of (**a**) Pd, (**b**) Ni, and (**c**) PdNi binary alloy NPs and their responding particle size distribution (**d**–**f**).

### 2.2. The Dispersion of Pd and Ni Species on CeO$_2$ of PdNiO-H Sample

The dispersion of Pd and Ni species on CeO$_2$, using the self-assembly approach, was explored through XRD and TEM characterizations. For comparison, a CeO$_2$-supported PdNi catalyst prepared via the impregnation method was also studied. Figure 2b shows the isothermal adsorption curves and pore size distribution of CeO$_2$, and the type of IV hysteresis loop indicates the presence of mesoporous structures within the bulk phase of CeO$_2$, and the pore size distribution predominantly centers on 8 nm, which proves suitable for alloy particles to enter the aperture structure and form a uniform distribution. Upon supporting Pd, Ni, and PdNi, the surface of CeO$_2$ decreased from 154.7 cm$^2 \cdot$g$^{-1}$ to 120.4, 121.7, and 120.2 cm$^2 \cdot$g$^{-1}$, respectively (Table 1). This indicates that some of the metal nanoparticles have partially entered the pores of CeO$_2$, including the impregnated catalyst. The prepared CeO$_2$ reveals the fluffy structure from the SEM image in Figure 3c. The TEM image confirms the existence of mesoporous structures in CeO$_2$, as evidenced by the bright streak in Figure 2d for the torn spheroplast. Tang et al. [22] proved that the mesoporous

structures of $CeO_2$ with a large surface area can improve the dispersion of NiO. Therefore, it can be considered that the synthesized $CeO_2$ is available to disperse Pd and Ni species.

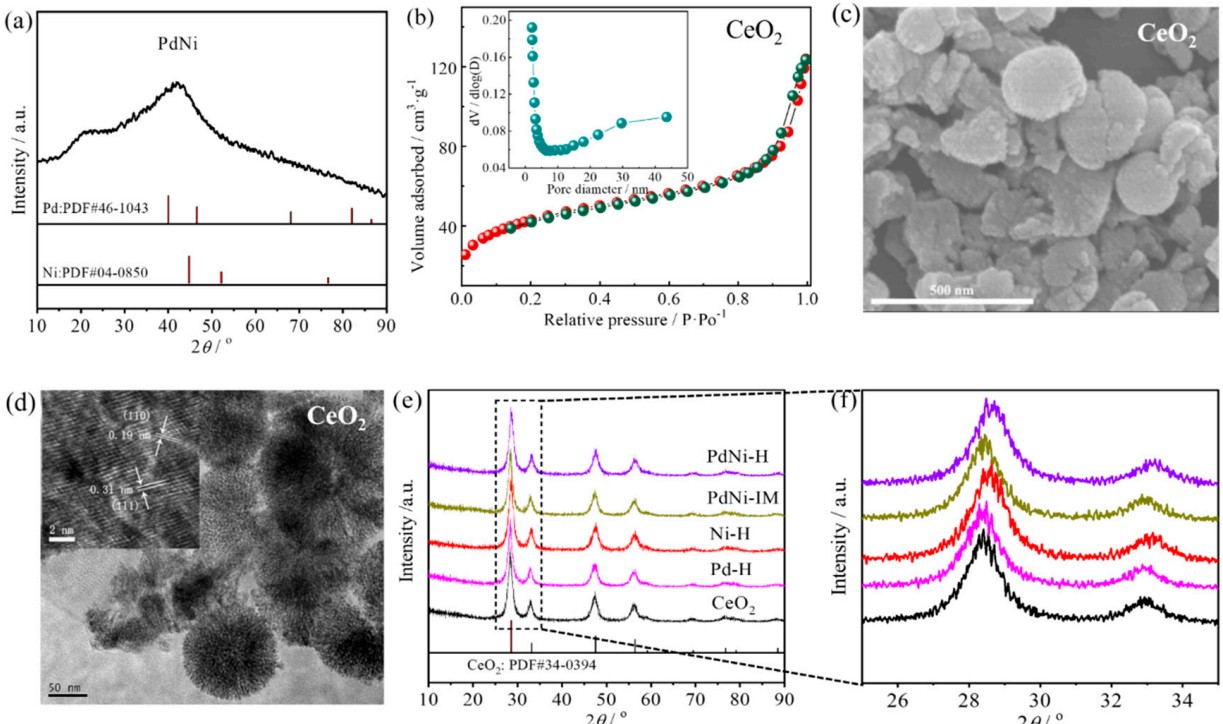

**Figure 2.** (**a**) XRD pattern of samples PdNi NPs; (**b**) isothermal adsorption and desorption curves and pore size distribution of $CeO_2$; (**c**) SEM and (**d**) (HR)TEM of $CeO_2$; (**e**) XRD patterns of $CeO_2$, Pd-H, Ni-H, PdNi-IM, and PdNi-H; and (**f**) corresponding profiles of selected degree between 25° and 35°.

**Table 1.** Lattice constant, crystallite sizes ($d_{XRD}$), and BET surface areas ($S_{BET}$) of the related samples.

| Sample | Lattice Constant (nm) [a] | $S_{BET}$ (m²/g) |
| :---: | :---: | :---: |
| $CeO_2$ | 0.05444 | 154.7 |
| Ni-H | 0.05414 | 121.7 |
| Pd-H | 0.05437 | 120.4 |
| PdNi -H | 0.05396 | 120.2 |
| PdNi-IM | 0.05429 | 121.8 |

[a] Calculated by the XRD results.

From the XRD patterns in Figure 3e, the $CeO_2$, Pd-H, Ni-H, PdNi-IM, and PdNi-H samples show the main peaks at $2\theta$ = 28.55°, 33.08°, 47.48°, and 56.33° can be assigned to the (111), (200), (220), and (311) reflections for $CeO_2$ (ICDD-JCPDS Card No. 34-0394), respectively, without responding diffraction peaks related to Pd or Ni species, which is related to the uniform dispersion of metal nanoparticles. The magnified profiles between 25° and 35°, as shown in Figure 2f, reveal shifts to higher diffraction angles after the introduction of Pd, Ni, and alloy nanoparticles. This suggests ion doping within the lattice of $CeO_2$. The maximum degree of shift observed in PdNi-H indicates a higher dispersion of PdNi nanoparticles compared to that in the PdNi-IM catalyst. Zou et al. reported that the metallic ion could be doped in the lattice $CeO_2$ for the strong interaction between the support and metal oxide, which would result in lattice distortion in $CeO_2$ [23]. The lattice parameters of $CeO_2$ were calculated through X-ray line broadening of all the diffractions, and using the crystallographic formula for the cubic crystal system, the lattice distance (d) values of the (111) plane of $CeO_2$ can be calculated; the results are illustrated in Table 1. Generally, the doped substrate will lead to a change in the lattice constant for the lattice distortion of $CeO_2$. The ion radii of $Pd^{2+}$ (r = 0.086 nm) and $Ni^{2+}$ (r = 0.069 nm) are much

smaller than $Ce^{4+}$ (r = 0.110 nm), which would lead to a decrease in the lattice constant of $CeO_2$. The lattice constants of $CeO_2$-supported samples shift to a lower quantitative value compared to the pure ceria and follow the sequence $CeO_2$ (0.05444 nm) > Pd-H (0.05437 nm) > PdNi-IM (0.05429 nm) > Ni-H (0.05414 nm) > PdNi-H (0.05396 nm). The increased change in the lattice constant of Ni-H sample may be related to the larger amount of supported Ni than the Pd molar amount in the Pd-H sample for the molar ratio of Pd: Ni = 1:5. Meanwhile, it is obvious that the PdNi-H shows the minimum value in lattice constant and the negligible change for PdNi-IM, indicating the maximum amount of $Pd^{2+}$ and $Ni^{2+}$ doped into the lattice of $CeO_2$ in the PdNi-H catalyst. According to Pal et al. [24] and Lian et al. [25], the Ce-O-Pd-O or Ce-O-Ni-O structures could form in the presence of metallic oxide after the doping of ion.

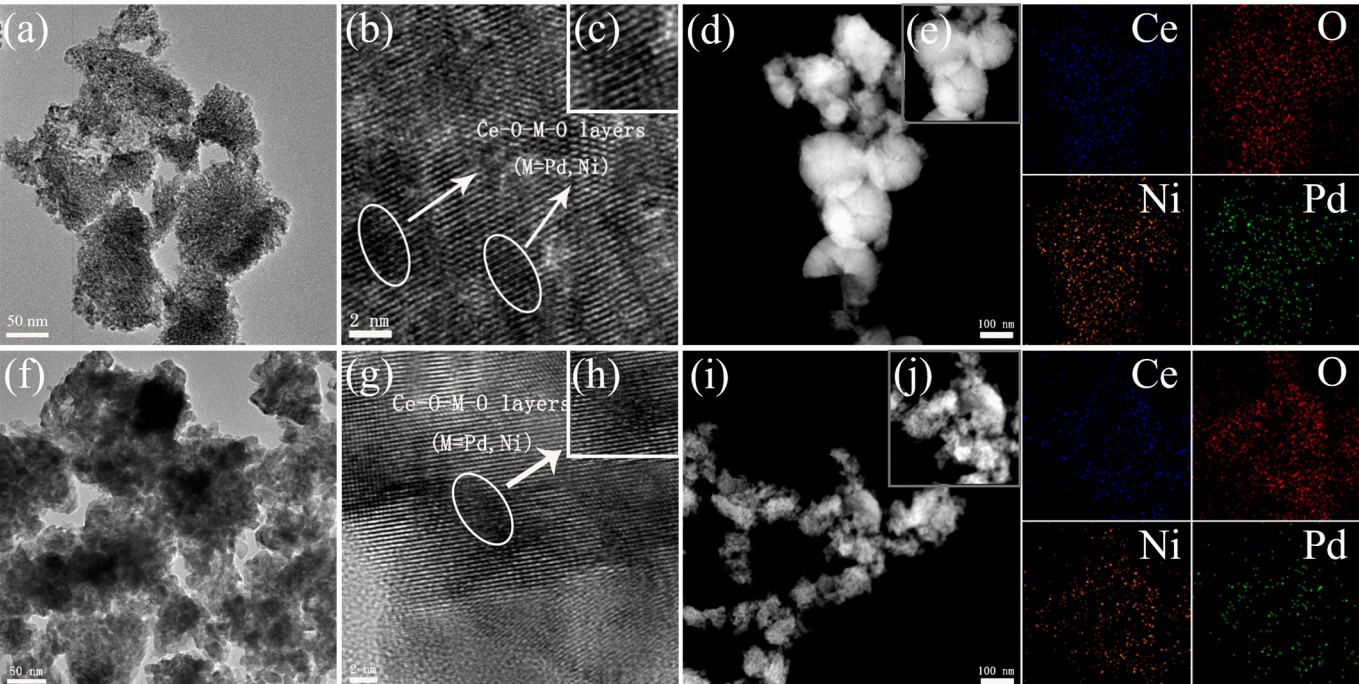

**Figure 3.** TEM and HR-TEM images of PdNi-H (**a–c**) and PdNi-IM (**f–h**)) samples in different magnification. Corresponding STEM HAADF and EDS mapping images of the selected region (**d,e,i,j**).

To underscore the superiority of the binary metal catalyst prepared through the self-assembly method, both the PdNi-H and PdNi-IM catalysts underwent TEM characterization and energy-dispersive X-ray spectroscopy (EDS) testing. From the TEM images of the two catalysts in Figure 3a,f, there are no obvious Pd or Ni species that can be observed on the support. The overlap of lattice fringe observed from the HRTEM image of Figure 3b,c,g,h can be attributed to the formation of the Ce-O-Pd bond, which is consistent with the XRD results and conducive to the stability of the metal nanoparticles. From the STEM images by high-angle annular dark-field mode in Figure 3d,i, we can see that there are no bright spots derived from metal nanoparticles, which can be ascribed to the high dispersion of the Pd and Ni species. Moreover, from the energy-dispersive X-ray spectroscopy (EDS) mappings (form Figure 3e,j) images, the well-matched distribution of Pd, Ni, O, and Ce elementals in the selected region for PdNi-H indicates the close contact between Pd and Ni species, and they are evenly dispersed. Meanwhile, the Pd and Ni for the PdNi-IM catalyst show mild aggregation for the presence of uneven bright specks.

*2.3. Reducibility of CeO$_2$ Supported Bimetallic Oxide Samples*

To obtain redox property information about pure and supported $CeO_2$ samples, the $H_2$-TPR experiments of $CeO_2$, Ni-H, Pd-H, PdNi-H, and PdNi-IM catalysts were performed,

and the profiles are shown in Figure 4. The pure $CeO_2$ sample shows a broad reduction peak centered at 609 °C, which is assigned to the reduction of $Ce^{4+}$ to $Ce^{3+}$ situated at the surface [26]. However, the TPR profiles of the supported samples show two distinctive reduction peaks, and the reduction temperatures all shift to lower temperature regions as compared to pure $CeO_2$. For the Pd-H sample, the reduction peaks at 145 °C and 246 °C can be assigned to the surface and bulk PdO. And for the Ni-H reduction process, the peaks at 326 °C and 388 °C are attributed to the reduction of NiO species with weak and strong interaction with $CeO_2$ supports, respectively. When compared with the monometallic catalyst, it is worth mentioning that the reduction peaks of bimetallic catalysts shift to a lower-temperature section. PdNi-IM shows a broad peak centered at 118 °C, with a shoulder peak at 151 °C, much lower than the Ni-H sample, indicating that the addition of Pd can effectively promote the reduction of NiO. When it comes to the PdNi-H sample, a sharp peak located at 104 °C and the shoulder peak at 122 °C are present in the reduction profile, much lower-temperature regions compared to the PdNi-IM sample, reflecting that the PdNi species show a stronger redox capacity and that the interaction between Pd and Ni species is stronger than that in PdNi-IM sample [27]. The obvious sharp peak represents the reduction of surface Pd and Ni species, and the other one is caused by the bulk NiO and PdO. Meanwhile, the narrow peak spacing indicates the uniform existential form of NiO, which benefits from the closer contact of PdO and NiO derived from the PdNi binary alloy. The $H_2$ consumption at lower temperatures signifies the greater number of surface Pd and Ni species, related to the better dispersion of the PdNi nanoparticles. According to Pudukudy [28], the interactive effects of Ni-Pd bimetallic species could be responsible for the high catalytic activity. In addition, the absence of reduction peaks in supported samples that respond to a $Ce^{4+}$ reduction can be related to the presence of Pd or Ni species.

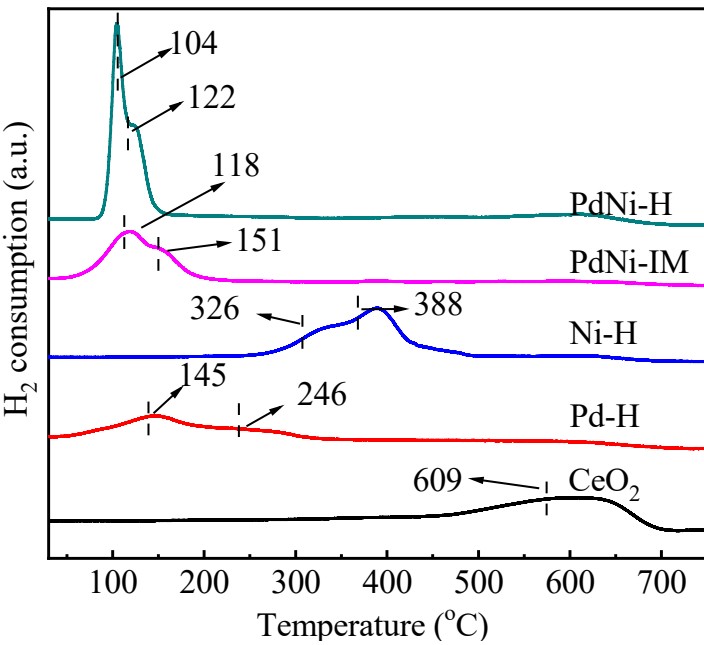

**Figure 4.** $H_2$-TPR profiles of $CeO_2$, Pd-H, Ni-H, PdNi-IM, and PdNi-H samples.

### 2.4. Oxygen Vacancy of $CeO_2$ Supported Samples

The oxygen vacancy plays a pivotal role in stimulating oxygen to generate reactive oxygen species. Raman spectra and X-ray photoelectron spectroscopy (XPS) characterization were carried out to investigate the surface oxygen vacancy of the synthesized hydrophobic $CeO_2$ and supported $CeO_2$ samples for their important role in oxygen activation. Figure 5a illustrates the Raman spectra with a typical region representing the defective structure of $CeO_2$. Notably, all catalysts show two distinct bands at 460 and 588 $cm^{-1}$ and three unobservable bands at 257, 830, and 1160 $cm^{-1}$. The bands at 257 and 460 $cm^{-1}$ are ascribed to

the second-order transverse acoustic (2TA) and Raman active F2g mode vibration of $CeO_2$ fluorine lattice (space group $O_{5h}$), in which the latter could be identified as a symmetric breathing mode of oxygen atoms around the cerium ions [29], and the band at 588 $cm^{-1}$ is attributed to oxygen vacancies, called the defect-induced (D) mode [29–32]. The D mode and the bands at 258 and 1160 $cm^{-1}$ arise from a combination of the $A_{1g}$, $E_g$, and $F_{2g}$ vibrational modes of the $CeO_2$ lattice. It has been reported that the value of $I_{588}/I_{460}$ is positively correlated with the presence of oxygen vacancy [32]. To observe the difference between each catalyst, the magnified profiles of this region are exhibited in Figure 5b. Compared to the pure $CeO_2$, the supported catalysts show a blue-shift trend, except for the Pd-H sample, which is caused by the substitution of $Ce^{4+}$ by $Pd^{2+}$ or $Ni^{2+}$ in the ceria structure, leading to weaker force constants for the Ce-O bond. Hence, the maximum incorporation of metallic ions into PdNi-H samples results in the most shifted wave number value, and it ranks second for the PdNi-IM sample. In addition, the relative results of oxygen vacancy were obtained by calculating the value of $I_{588}/I_{460}$, as shown in Figure 5b. They are 0.009, 0.01, 0.012, 0.034, and 0.042 for $CeO_2$, Pd-H, Ni-H, PdNi-IM, and PdNi-H, indicating that the oxygen vacancy concentration reaches a maximum for PdNi-H [33]. Interestingly, low-intensity bands at 830 $cm^{-1}$ disappear in the Ni-H, PdNi-IM, and PdNi-H samples in Figure 5a. According to Rotaru [32], the band is related to the O-O stretching vibration of peroxide ($O_2^{2-}$) with different degrees of defect aggregation, whose disappearance can be explained by the reduction of surface peroxide ($O_2^{2-}$) after the doping of $Pd^{2+}$ or $Ni^{2+}$. In turn, peroxides convert into lattice oxygen ($O^{2-}$) and oxygen vacancy ($V_o$), which results in a slight increase in the intensity band at 1160 $cm^{-1}$ and the $I_{588}/I_{460}$ value.

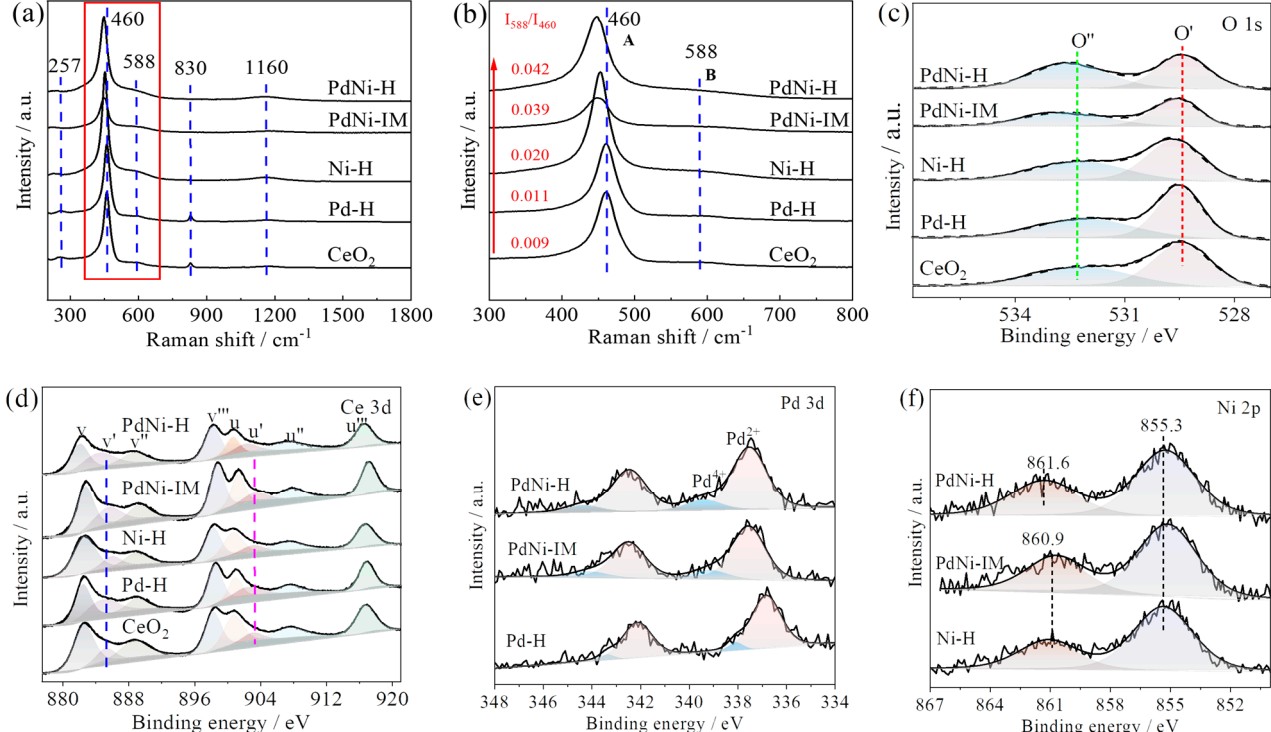

**Figure 5.** (**a**) Visible Raman spectra of synthesized $CeO_2$, Pd-H, NiO-H, PdNi-IM, and PdNi-H samples and (**b**) their responding evolution of the $F_{2g}$ (A) and D-line (B) Raman mode. XPS spectra of (**c**) O 1s and (**d**) Ce 3d for above samples. (**e**) Pd 3d spectra for Pd-H, PdNi-IM, and PdNi-H catalysts. (**f**) Ni 2p spectra for Ni-H, PdNi-IM, and PdNi-H samples.

XPS characterization was combined here to determine the oxygen vacancy exposed on these catalysts, which is assumed to be an adequate descriptor for the oxygen activation in methane combustion reactions. Figure 5c–f show the O1s, Ce3d, Pd3d, and Ni2p spectra of related samples. In the O1s spectra, the main peak (O′) at 529.5 eV can be assigned to

the lattice oxygen ($O^{2-}$) in ceria, while the peak ($O''$) at 532.2 eV corresponds to surface-adsorbed oxygen species (e.g., $O_2^-$, $O_2^{2-}$, or $O^-$) due to the substitution of metal ions [34], as this plays an important role in the deep oxidation that takes place in methane combustion. After deconvolution and multiple-peak separation, the quantitative value of $O''/(O'' + O')$ was calculated to obtain the relative amount of surface active oxygen, and the results are summarized in Table 2. It is noted that the increased value of $O''/(O'' + O')$ in the Pd-H, Ni-H, PdNi-IM, and PdNi-H samples can be attributed to the rising amount of active oxygen, indicating that bimetallic doping is more conducive to forming reactive oxygen species, especially the highly dispersed bimetallic oxides.

**Table 2.** The quantitative analysis results according to the XPS spectra after fitting responding peaks of O1s and Ce3d.

| Catalysts | $O''/(O'' + O')$ [a] | $Ce^{3+}$ (%) [b] | $Pd^{4+}/Pd^{2+}$ [c] |
|---|---|---|---|
| $CeO_2$ | 0.42 | 12.11 | - |
| Pd-H | 0.43 | 14.92 | 0.10 |
| Ni-H | 0.47 | 17.75 | - |
| PdNi-IM | 0.48 | 21.89 | 0.17 |
| PdNi-H | 0.53 | 30.59 | 0.20 |

[a] $O'$ corresponds to the lattice oxygen, and $O''$ represents for the adsorbed oxygen. [b,c] Calculated by deconvoluting the multiple peaks.

Simultaneously, the Ce3d spectra were quantitatively estimated by deconvoluting the multiple peaks to further confirm the relative quantity of oxygen vacancy. As shown in Figure 5d, the peaks labeled at u and v series represent the Ce $3d_{3/2}$ and $3d_{5/2}$ spin–orbit components, respectively. The four main peaks at 882.7, 885.3, 888.7, and 898.6 eV belong to Ce $3d_{5/2}$ and are defined as v, v′, v′, and v′′′ components, while the Ce $3d_{3/2}$ located at 900.9, 903.2, 907.8, and 916.8 eV correspond to the u, u′, u′′, and u′′′ components [35–37]. According to Zhang [33], the characteristic peaks of $Ce^{4+}$ ions are denoted by v, v′′, v′′′, u, u′′, and u′′′, whereas the characteristic bands at v′ and u′ are featured to $Ce^{3+}$ ions. The ratio of $Ce^{3+}$ in the whole Ce concentration of these samples can be calculated from the area of the peak, and the results of $Ce^{3+}/(Ce^{3+} + Ce^{4+})$ are illustrated in Table 2 and in the following equation:

$$Ce^{3+}(\%) = \frac{S_{u'} + S_{v'}}{\sum(S_{u'} + S_{v'})} \times 100 \tag{1}$$

The content of oxygen vacancies exhibits a linear proportionality with the amount of $Ce^{3+}$ for the charge imbalance and unsaturation of chemical bonds after the $Pd^{2+}$ and $Ni^{2+}$ doping of the lattice of $CeO_2$. The formal equation is as follows: $Ce_4O_8 \rightarrow Ce_4O_7 + O_v$ [30,38,39]. From Table 2, we can see that the calculated concentration of $Ce^{3+}$ increases by a considerable amount with the following order: $CeO_2$ (12.11%) < Ni-H (14.92%) < Pd-H(17.75%) < PdNi-IM (21.89%) < PdNi-H (30.59%). Compared to the pure $CeO_2$, the $Ce^{3+}$ in the PdNi-H sample almost increased three times, and that is two times for the PdNi-IM sample. Therefore, the bimetallic oxides are more likely to promote the formation of oxygen vacancy, especially the two with a strong interaction, which is also confirmed by the Raman result.

The charge state of palladium species on the surface of $CeO_2$ can be obtained via an analysis of the Pd3d doublet line (Figure 5e). The binding energy of Pd ($3d_{5/2}$) located at 336.8 eV can be assigned to $Pd^{2+}$ stabilized in PdO; the elevated value of $Pd^{4+}$ ($Pd3d_{5/2}$) in the Pd-H, PdNi-IM, and PdNi-H samples is caused by the localization of individual palladium ions in the ceria lattice [30], in which the last one shows the maximum $Pd^{4+}/Pd^{2+}$ value, indicating the greater doping amount of metal ions in $CeO_2$. Figure 5f shows $Ni2p_{3/2}$ for the Ni-H, PdNi-IM, and PdNi-H samples, with a main peak centered at 855.2 eV assigned to $Ni^{2+}$. In addition, a broad peak at 860.9 eV is assigned as the shake-up satellite peak of $Ni^{2+}$ (NiO) and $Ni^{3+}$ ($Ni_2O_3$), respectively [40], consistent with the TPR analysis. Apparently, only the PdNi-H catalyst presents higher binding energy (851.6 eV) for the

$Ni_2O_3$ peaks after curve deconvolution, indicating a greater electron transfer from Ni to Pd species [17].

*2.5. The Catalytic Performance of CeO$_2$ Supported Bimetallic Catalysts in Methane Combustion Reaction*

The influence of bimetallic doping on the catalytic activity of $CeO_2$-supported catalysts for the methane combustion reaction was investigated here at GHSV of 30,000 h$^{-1}$, and the temperature ranged between 250 and 600 °C with a gas mixture comprising a mixture of $CH_4$:$O_2$:$N_2$ with a volume ratio of 1:4:95, guaranteeing the full combustion of methane.

On the basis of the Pd/Ni molar ratio of 1:5, 0.05 wt%, 0.25 wt%, 0.5 wt%, 0.75 wt%, and 1 wt%, PdNi binary metal catalysts supported by $CeO_2$ (based on the Pd mass load) were synthesized, and the catalytic performance was studied. As shown in Figure 6a, with the increase in alloy load, the catalytic conversion curves of methane shift towards a low temperature until the PdNi alloy loading capacity increases to 0.5 wt% in terms of the mass fraction of Pd. The ignition temperature of methane drops from 384 to 282 °C when the loading of Pd increases from 0.05 wt% to 0.5 wt%. In addition, the complete combustion temperature of methane (387 °C) for the PdNi-H catalyst with 0.5 wt% PdNi loading shows absolute superiority compared to lower Pd loading catalysts. When the temperature continues to rise to 600 °C, the PdNi-H catalyst does not show a deactivation phenomenon, which indicates that the catalyst has a certain stability at high temperatures. When the Pd loadings increase to 0.75 wt% and 1 wt%, the activity of the PdNi-alloy-supported catalysts decreases; thus, it can be speculated that the aggregation can be caused by a large amount of PdNi alloy supporting. We further studied the effect of the Pd: Ni ratio on the activity of methane combustion. Figure 6b shows the methane conversion temperature at the 10%, 50% and 90% conversion (defined as $T_{10}$, $T_{50}$, and $T_{90}$, respectively) of the $CeO_2$-supported PdNi alloy catalysts with the Pd: Ni ratio for 1:2.5, 1:5, and 1:10. The plot of $T_{10}$, $T_{50}$, and $T_{90}$ versus Pd: Ni ratio demonstrates a volcano-shape relationship, and the PdNi-H catalyst with 1:5 molar ratio shows an outstanding $T_{10}$ of 282 °C, $T_{50}$ of 351 °C, and $T_{90}$ of 387 °C, outperforming all other candidates. This result further stresses that a 1:5 ratio for Pd: Ni is the optimal supported concentration. Figure 6c is the temperature-dependent $CH_4$ conversion of $CeO_2$, Pd-H, Ni-H, PdNi-IM, and PdNi-H catalysts. It can be obviously seen that the initial burning ($T_{10}$) and complete oxidation ($T_{90}$) temperature of $CH_4$ for PdNi-H catalyst drop by 92 and 212 °C compared to the Pd-H and Ni-H catalysts, while the complete oxidation temperature lowers to 200 and 202 °C, indicating that the bimetallic synergy could promote the catalytic performance in methane combustion. Meanwhile, the IM sample exhibits poor catalytic activity for the $T_{10}$ and $T_{90}$ at 344 and 520 °C, much higher than that for the PdNi-H sample. To explore the inherent differences in the catalytic activity of these catalysts, kinetic experiments were conducted, and the related activation energy results are shown in Figure 6d. The activation energies of $CeO_2$, Ni-H, Pd-H, PdNi-IM, and PdNi-H are 119.8, 83.5, 69.5, 59.2, and 47.5 kJ·mol$^{-1}$, respectively. The lowest activation energy for PdNi-H illustrates the enhanced activity of its dual reactive sites, as is consistent with the highest catalytic activity, especially at low temperatures. In terms of the Pd loading and catalytic activity, the activity of PdNi-H is comparable to that of the representative Pd-based binary metal catalysts reported in the past decade (Figure 6e) and in Table 3. Moreover, the PdNi-H catalyst also exhibits outstanding stability with a negligible change in $CH_4$ conversion over a 50 h time-on-stream experiment at 350 °C (Figure 6f).

Based on the above results, we speculate on the mechanism of the catalytic conversion of methane (shown in Figure 7).

In the catalytic process, methane dissociates at the active site ($Pd^{4+}$ and $Ni^{3+}$) and forms methyl and methylene species. These species interact with removable lattice oxygen ($O^{2-}$) on the surface of the catalyst and are fully oxidized to carbon dioxide and water, leading to the generation of oxygen vacancies ($O_v$) and $Ce^{3+}$. Gaseous oxygen preferentially adsorbs on the oxygen vacancy to replenish oxygen species. The defect healing process involves the re-oxidation of $CeO_2$ and metal irons and the formation of surface superoxide

species ($O_2^-$) through a direct interaction of $O_2$ with low-coordinated $Ce^{3+}$ ions. These superoxide species act as a stronger oxidant to react with methane, while the lattice oxygen in the $CeO_2$ remains intact.

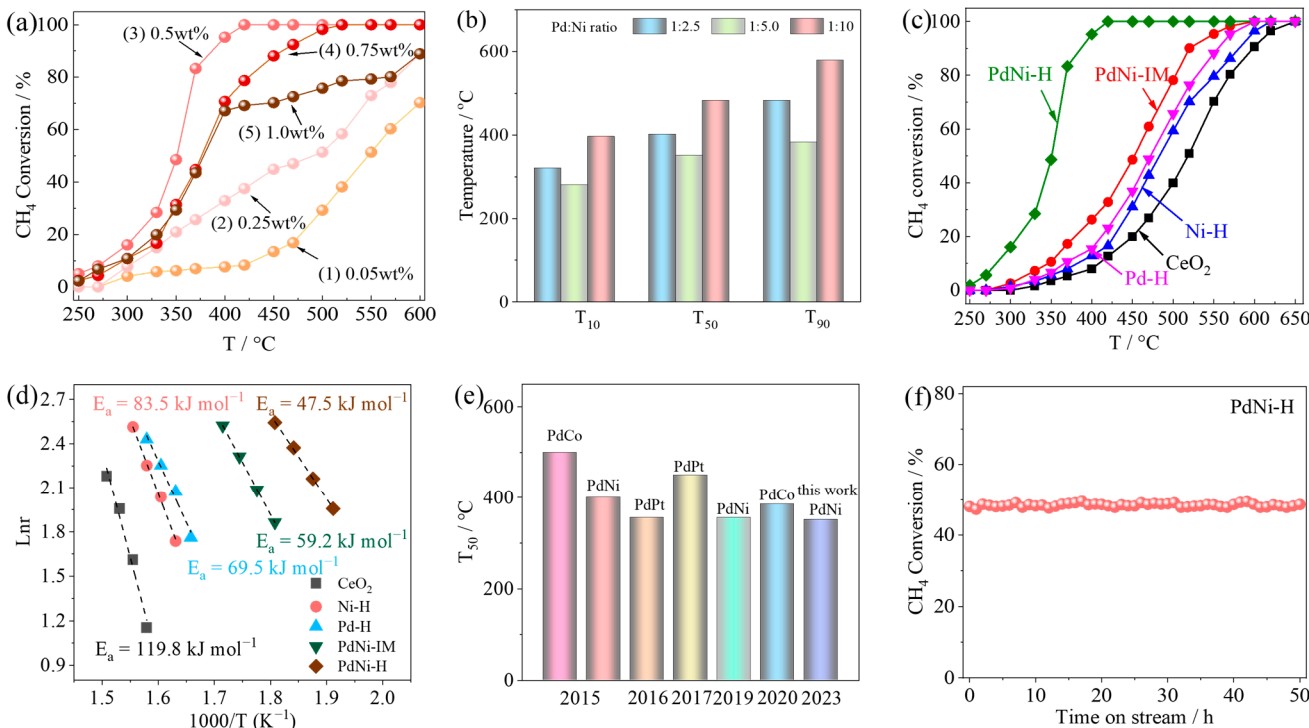

**Figure 6.** (**a**) Methane conversion heating curves against temperature over PdNi-H catalysts with various PdNi (Pd:Ni = 1:5 in molar amount) loading amounts (from 0.05 wt.% for Pd to 1.0 wt.%). (**b**) The $T_{10}$, $T_{50}$, and $T_{90}$ of PdNi-H catalysts with different Pd:Ni ratios at the same 0.5 wt.% Pd loading. (**c**) Temperature-dependent $CH_4$ conversion of $CeO_2$, Pd-H, Ni-H, PdNi-IM, and PdNi-H catalysts. (**d**) Steady-state, differential reaction rates for $CH_4$ oxidation. (**e**) Comparison of the $T_{50}$ values of representative binary metal oxide catalyst systems [9,15,41–44]. (**f**) The stability test of the PdNi-H catalyst at 350 °C. (All of the above reactions were conducted at the conditions of 1 vol.% $CH_4$, 5 vol.% $O_2$, balanced with He, and GHSV = 30,000 $h^{-1}$).

**Table 3.** The comparison of Pd loading, feed concentration, GHSV, and $T_{50}$ for the reported works and this work.

| Catalysts | Pd wt.% | Feed Concentration | GHSV ($h^{-1}$) | $T_{50}$ (°C) | Reference |
|---|---|---|---|---|---|
| PdCo/$Al_2O_3$ | 0.5 | 0.4%$CH_4$/10%$O_2$/$N_2$ | 300,000 | 500 | [9] |
| PdNi/$Al_2O_3$ | 0.029 | 4100 ppm methane, 5 mol % water | 3128 | 400 | [15] |
| PtPd/ $MnLaAl_{11}O_{19}$ | 0.67 | 5%$CH_4$ in air | 10,000 | 355 | [41] |
| PtPd/$Al_2O_3$ | 0.5 | 0.5%$CH_4$/2.0%$O_2$/Ar | 200,000 | 450 | [42] |
| PdNi@Hal | 1 | 1%$CH_4$/20%$O_2$/Ar | 72,000 | 355 | [43] |
| Pd/6CoDEG/OMA | 0.5 | 1%$CH_4$/10%$O_2$/$N_2$ | 30,000 | 385 | [44] |
| PdNi/H | 0.5 | 1%$CH_4$/5%$O_2$/$N_2$ | 30,000 | 351 | This work |

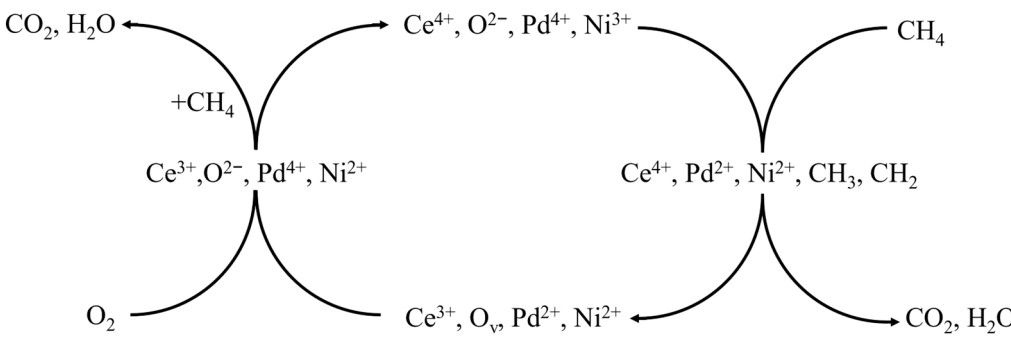

**Figure 7.** Catalytic cycle for $CH_4$ oxidation on PdNi-H catalyst.

## 3. Experimental Section

### 3.1. Preparation of Catalysts

3.1.1. Synthesis of PdNi-H Catalysts

The synthesis strategy to prepare catalysts was like the following route presented in Figure 8. All reactants were used without further purification.

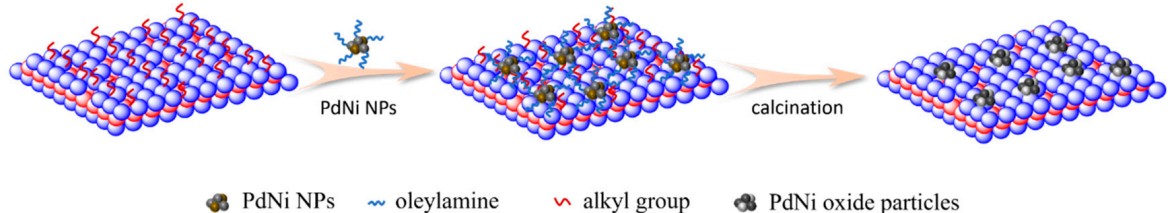

**Figure 8.** Schematic of preparation of $PdO$-$NiO/CeO_2$-H sample.

The typical procedures are as follows: 0.0625 g of nickel (II) acetylacetonate ($Ni(acac)_2$, Sigma Aldrich, St. Louis, MO, USA, 95%) and 0.0152 g of palladium (II) acetylacetonate ($Pd(acca)_2$, Sigma Aldrich, 98%) were mixed in 10 mL of oleylamine (OAm, Sigma Aldrich, ≥98%). The reaction was performed in 3-neck-flask equipment connected to a surge flask under a $N_2$ atmosphere. Under continuous stirring, the mixture was heated to 60 °C until completely dissolved, and then 1 mL of trioctylphosphine (TOP, Sigma Aldrich, St. Louis, MO, USA, 90%) was injected. When the mixed solution in stirring was heated to 240 °C, the light green solution changed to dark green rapidly and remained that was for 45 min; then, the dark solution was naturally cooled to room temperature. Ethanol was added as a precipitate to obtain PdNi nanoparticles. The products were separated by means of centrifugation and then dissolved in 20 mL of cyclohexane for the next step. The molar ratios of Pd/Ni were 1/2.5, 1/5, and 1/10.

Then, 3.47 g of cerium nitrate hexahydrate ($Ce(NO_3)_3 \cdot 6H_2O$, Aladdin, 99%) was dissolved in 50 mL of benzyl alcohol (Sinopharm Chemical Reagent Co., Shanghai, China, 99%), and stirred for 10 min at 100 °C to remove the water. The solution was kept in a refluxing device at 140 °C for 12 h, precipitated by adding ethanol, and then it was washed with ethanol three times. The deep yellow product was air-dried at 60 °C overnight. The obtained solution was named polyporous $CeO_2$.

A total of 1.06 g of the obtained polyporous $CeO_2$ was added to 20 mL of cyclohexane solution containing PdNi alloy nanoparticles, stirred at room temperature until it looked like a gel, air-dried at 60 °C for 6 h, and then calcinated at 450 °C for 2 h. The catalyst was labeled PdNi-H; the "H" stands for metallic oxides derived from the PdNi alloy synthesized through high-temperature pyrolysis.

The polyporous $CeO_2$-supported Pd and Ni species were prepared through the same procedure, denoted as Pd-H and Ni-H, respectively.

### 3.1.2. Synthesis of PdNi-IM

The PdNi-IM catalyst was prepared with an impregnation method, using the Ni(acac)$_2$ (0.25 mmol) and Pd(acca)$_2$ (0.05 mmol) as precursors. The two precursors were dissolved in 20 mL of cyclohexane, and 1.06 g of polyporous CeO$_2$ was then added. The mixture was then stirred at room temperature until it looked like a gel, air-dried at 60 °C for 6 h, and calcinated at 450 °C for 2 h. The catalyst was labeled PdNi-IM.

### 3.2. Catalytic Performance Test

The catalytic performances of all samples were investigated using a mixture gas consisting of methane, oxygen, and nitrogen with a volume ratio of 1/4/95 over the temperature range from 250 to 650 °C with a gas hourly space velocity (GHSV) of 30,000 mL·g$_{cat}$$^{-1}$·h$^{-1}$. The reaction was implemented in a fixed bed quartz reactor (internal diameter = 5 mm) at atmospheric pressure. A total of 100 mg of fresh catalysts (40−60 mesh sizes) was filled into the center of the quartz tube, and the upstream and downstream of the catalysts were fixed with quartz cotton. The quartz tube was placed in a furnace connected to an automatic thermometer (Yudian Technology Co., Ltd. AI−708PFKSL2, Xiamen, China). The gas products were analyzed by gas chromatograph (Haixin instrument Co., Ltd., 9790 GC, Shanghai, China) with a TCD. Prior to the tests, 100 mg of the catalyst was heated to 250 °C, with a rate of 10 °C/min, from room temperature under a N$_2$ atmosphere, and then the gas reactant mixture was passed into the quartz reactor, with a total flow rate of 50 mL·min$^{-1}$. The CH$_4$ conversion ($X_{CH4}$) was calculated as follows:

$$X_{CH_4} = \frac{C_{CH4(in)} - C_{CH4(out)}}{C_{CH4(in)}} \times 100\% \tag{2}$$

where $X_{CH4}$ represents the CH$_4$ conversion, $C_{CH4(in)}$ denotes the concentration of methane in the feed, and $C_{CH4(out)}$ refers to the concentration of methane at the reactor outlet.

### 3.3. Characterization Techniques

Powder X-ray diffraction patterns of the samples were obtained from a Panalytical X'pert PRO diffractometer (Philips, Amsterdam, Netherlands). The samples were scanned from 10 to 90° 2$\theta$. Cu K$_\alpha$ radiation ($\lambda$ = 0.15406 nm) generated at 40 kV and 30 mA was used as the X-ray source.

H$_2$-temperature programmed reduction (H$_2$-TPR) was carried out in a homemade fixed-bed reactor to study the interaction between Pd and Ni species. A total of 100 mg of catalyst sample was put in the center of the quartz reactor plugged with ceramic wool. The sample was first pretreated in a flow of pure O$_2$ (40 mL·min$^{-1}$) at 500 °C for 30 min, oxidizing the metal phase thoroughly. Then, the H$_2$-TPR experiment was performed by heating the catalyst sample from 20 to 900 °C at a rate of 10 °C·min$^{-1}$ in a 5 vol% H$_2$/Ar gas mixture with a total rate of 25 mL·min$^{-1}$, and the H$_2$ uptake signal was measured using a TCD.

Transmission electron microscopy (TEM) images were taken on a Tecnai G2 F20 (FEI, Hillsboro, OR, USA) operating at 200 kV to observe the particle morphology of the sample, and energy dispersive X-ray (EDX)-mapping was also taken to confirm the distribution of elements in the selected area of the samples.

X-ray photoelectron spectroscopy (XPS) analysis was achieved on Quantum 2000 scanning ESCA microprobe instrument (Physical Electronics, Minneapolis, Minnesota, USA) with Al K$\alpha$ irradiation at the pressure of 5.0 × 10$^{-9}$ mbar. The spectra were corrected by C1s labeled peak at 284.5 eV. The detailed operating steps are as follows: the freshly prepared catalyst was compressed into a compacted slice and positioned in a self-designed in situ treatment cell. It was then treated with nitrogen for 1 h. Subsequently, the slice was transferred to a vacuum chamber for XPS measurements in its pristine state.

The surface areas of the samples were derived from the N$_2$ adsorption and desorption isotherms at −196 °C, using automated Micromeritics Tristar 3000 equipment (Norcross,

Gwinnett Country, GA, USA). Before analysis, the samples were degassed at 300 °C for 3 h under vacuum.

Raman characterization was carried out to verify the surface oxygen species of samples. The spectra were recorded on a Renishaw Invia Spectrometer Raman System 1000 with a confocal micro-Raman system, using a laser excitation line of 514 nm, and the output power was 5 mW with the 2000~200 $cm^{-1}$ spectra region.

### 4. Conclusions

In conclusion, we successfully prepared the PdNi-H catalyst by assembling the PdNi alloy on $CeO_2$. The bimetallic catalysts showed a synergistic effect on the methane's combustion performance. The XRD and TEM results demonstrated that metallic ions doped in the $CeO_2$ reduced lattice parameters, resulting in an increase in oxygen vacancy, which was further confirmed by the Raman spectra and XPS results. Moreover, the bimetallic catalysts exhibited better redox properties, which lower reduction temperatures for Pd and Ni species compared to the monometallic Pd-H and Ni-H catalysts, as evidenced by the $H_2$-TPR result. The PdNi-H catalyst outperformed the PdNi-IM catalyst prepared by the impregnated method, exhibiting the highest catalytic activity, particularly at lower temperatures. The superior performance can be attributed to the higher amount of oxygen vacancy derived from the doping of metallic ions in $CeO_2$ and its superior redox properties.

**Author Contributions:** Conceptualization, C.L. and Y.Y.; methodology, Y.Y.; investigation, Y.Y.; writing—original draft preparation, C.L., Y.Y. and M.T.; writing—review and editing, M.T., L.C., C.L., Y.Y., J.W. and A.N.A.; supervision, Y.Y. and A.N.A. All authors have read and agreed to the published version of the manuscript.

**Funding:** This work was supported by the National Natural Science Foundation of China (No. 22202041). The author Abdullah N. Alodhayb acknowledges Researchers Supporting Project (RSP2023R304), King Saud University, Riyadh, Saudi Arabia.

**Data Availability Statement:** All data used in the study appear in the submitted article.

**Conflicts of Interest:** The authors declare no conflict of interest.

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
