# Peer review of "Uniformly Dispersed Nano Pd-Ni Oxide Supported on Polyporous CeO2 and Its Application in Methane Conversion of Tail Gas from Dual-Fuel Engine"

_catalysts, doi:10.3390/catal14010024_

Round 1
Reviewer 1 Report
Comments and Suggestions for Authors
This paper presents interesting and useful results obtained using original preparation method and modern characterization techniques. Presentation of results is proper, discussion is adequate, conclusions are sound. However, it is to be brought to international standards before publishing.
1. Samples preparation for XPS studies is to be described, since surface properties are sensitive to contact with air or any liquid used for samples dispersion before supporting on samples holder in some procedures. In ideal case samples are to be pretreated in situ in XPS chamber in O2 followed by degassing at room temperature.
2. English is to be brought to standards with the help of native English speaker, some places requiring correction are marked in manuscript
So minor revision is required

Comments on the Quality of English LanguageEnglish is to be polished with the help of native English speaker
Author Response
Response to Reviewer 1 Comments
- Summary
Thank you for your thorough review of the manuscript. Please see the detailed responses below, and you can find the corresponding revisions/corrections highlighted in red in the resubmitted files using track changes. With these modifications, we trust that the revised paper aligns with the standards for publication in Catalysts.
- Point-by-point response to Comments and Suggestions for Authors
Comments and Suggestions for Authors
This paper presents interesting and useful results obtained using original preparation method and modern characterization techniques. Presentation of results is proper, discussion is adequate, conclusions are sound. However, it is to be brought to international standards before publishing.
Comments 1: Samples preparation for XPS studies is to be described, since surface properties are sensitive to contact with air or any liquid used for samples dispersion before supporting on samples holder in some procedures. In ideal case samples are to be pretreated in situ in XPS chamber in O2 followed by degassing at room temperature.
Response 1: Thank you for the valuable suggestions from the reviewer. We have now included a detailed description of the XPS sample preparation process, which has been added and highlighted in yellow within the Experimental section (Page 12, line 425).
Comments 2: English is to be brought to standards with the help of native English speaker, some places requiring correction are marked in manuscript, so minor revision is required.
Response 2: Thank you for pointing that out. The revised paper has undergone a thorough English check by a native English speaker, and corrections have been made in the highlighted yellow sections of the manuscript.
Reviewer 2 Report
Comments and Suggestions for Authors
The present article deals with the bimetallic (Pd, Ni) dispersed CeO2 catalysts for lean methane combustion. The research work is well-supported by the necessary characterization techniques. The role of redox properties of the catalysts is highlighted. However, the following comments need to be addressed for the acceptance.
Major comments:
1. What is the role of catalyst's hydrophobicity in lean methane combustion? The authors highlight an unimportant property in this work (in the title, results and discussion). Further, no studies are presented to prove hydrophobic nature of the catalyst.
2. Why the dying temperature was chosen as 60oC, when the solvent (cyclohexane) boiling temperature is high.
3. Line 216, 217: A metal ion can be substituted in the lattice of CeO2, not the metal oxide.
Authors claim that Pd and Ni exist in their oxide states (PdO and NiO). However, separate oxide peaks were not seen in XRD. Peak shift in XRD shows the possibility of ionic substitution in the lattice. Do Pd and Ni really exist in oxides, or are Pd and Ni ions substituted in Ce4+? or do both exist?
4. What is the elemental composition of Pd and Ni? Probably, this will help to understand the efficacy of lattice substitution.
5. Line 226, 227: How the lattice constants were estimated?
6. Figure 6 (c) and (f): The dotted lines are not representing the peak maxima. Clarity is missing. I also notice peak shift, authors need to explain the shift.
7. Line 409, 410: Literature comparison of activation energy or TOF, feed concentration, and GHSV is more appropriate than T50 comparison (as shown in Figure 7 (e)).
8. Line 267, 268: Why do the reduction peaks of bimetallic catalysts shift to low temperatures?
9. Explain the relations used in calculating apparent activation energy.
Minor Comments:
1. Line 98, 124: inconsistency with Ni precursor notation.
2. Line 198, 201: formatting issue
3. Line 183, 210: incorrect representation of JCPDS file
4. Line 254: Is it HAADF or HADDF? Explain the acronym.
5. Line 141: Correct the formula consistent to Figure 7. Conversion should be defined on molar flow basis. Based on chromatographs, the given formula is correct but should be defined scientifically. Give the equation numbering.
Author Response
Response to Reviewer 2 Comments
- Summary
Thank you for your thorough review of the manuscript. Please see the detailed responses below, and you can find the corresponding revisions/corrections highlighted in red in the resubmitted files using track changes.
- Point-by-point response to Comments and Suggestions for Authors
Comments and Suggestions for Authors
The present article deals with the bimetallic (Pd, Ni) dispersed CeO2 catalysts for lean methane combustion. The research work is well-supported by the necessary characterization techniques. The role of redox properties of the catalysts is highlighted. However, the following comments need to be addressed for the acceptance.
Major Comments:
Comments 1: What is the role of catalyst's hydrophobicity in lean methane combustion? The authors highlight an unimportant property in this work (in the title, results and discussion). Further, no studies are presented to prove hydrophobic nature of the catalyst.
Response 1: Thank you for your comment. In this manuscript, we have highlighted the hydrophobic nature of the CeO2 support, emphasizing its role in bonding with PdNi nanoparticles that carry oil-based protective groups. This bonding mechanism contributes to the high dispersion of PdNi species on the CeO2 support, ultimately promoting the exposure of active sites.
Comments 2: Why the dying temperature was chosen as 60oC, when the solvent (cyclohexane) boiling temperature is high.
Response 2: Thank you for your comment. The drying temperature of 60°C was selected to prevent rapid solvent volatilization, which could potentially lead to the blockage of the pores in the cerium dioxide carrier. The aim was to maintain the cerium dioxide in a fluffy state as much as possible. After overnight drying at 60°C, it ensured sufficient time for the complete volatilization of the solvent.
Comments 3: Line 216, 217: A metal ion can be substituted in the lattice of CeO2, not the metal oxide. Authors claim that Pd and Ni exist in their oxide states (PdO and NiO). However, separate oxide peaks were not seen in XRD. Peak shift in XRD shows the possibility of ionic substitution in the lattice. Do Pd and Ni really exist in oxides, or are Pd and Ni ions substituted in Ce4+? or do both exist?
Response 3: Thanks for pointing this out. Based on the results of XRD analysis, the variation in lattice parameters can be attributed to the incorporation of a small amount of Pd and Ni metal ions into the lattice of cerium dioxide. However, a substantial quantity of Pd and Ni is present on the surface of cerium dioxide in the form of oxides, which may go undetected due to the small particle size. This doping process is typically regarded as the substitution of metal ions for Ce4+, resulting in lattice distortion.
Comments 4: What is the elemental composition of Pd and Ni? Probably, this will help to understand the efficacy of lattice substitution.
Response 4: The catalyst preparation process is outlined as follows: initially, PdNi alloy nanoparticles are synthesized using a specific protective agent to prevent agglomeration. Subsequently, the alloy nanoparticles with the protective agent are dispersed onto the surface of the hydrophobic cerium dioxide, and the oily protective groups are eliminated through roasting. Finally, the PdNi alloy nanoparticles are dispersed on the cerium dioxide surface, leading to the formation of oxides. Consequently, we can infer that the high-temperature roasting process facilitates the migration of precious metal ions.
Comments 5: Line 226, 227: How the lattice constants were estimated?
Response 5: Thank you for the comment. Based on the XRD results, we have obtained the 2θ values of the diffraction peak corresponding to the (111) plane. The interplanar distance (d value) of the (111) plane for CeO2 can be calculated by substituting the θ values into the Bragg equation:
2d sinθ =nλ (1)
According to the crystallographic formula for the cubic crystal system, the relationship between interplanar distance and lattice parameters is as follows:
1/d2 =h2/a2+k2/b2+l2/c2 (2)
Here, d is the interplanar distance; a, b, and c are the length, width, and height values of a single CeO2 cell; are the Miller indices of the (111) plane along the x, y, and z axes. Given the characteristics of the cubic crystal system, we can accurately calculate the lattice parameters of cerium dioxide.
Comments 6: Figure 6 (c) and (f): The dotted lines are not representing the peak maxima. Clarity is missing. I also notice peak shift, authors need to explain the shift.
Response 6: Thanks for the suggestion. We have replaced the correct images in Figure 6c and f. As depicted in Figure 6f, a broad peak at 860.9 eV and 861.6 eV can be attributed to the shake-up satellite peaks of Ni2+ (NiO) and Ni3+ (Ni2O3), respectively. This occurrence may be a result of the synergistic interaction between Ni and the carrier.
Comments 7: Line 409, 410: Literature comparison of activation energy or TOF, feed concentration, and GHSV is more appropriate than T50 comparison (as shown in Figure 7 (e)).
Response 7: Thanks for the comment and suggestion. The comparison of T50 in the study of catalyst performance in methane combustion reactions provides a convenient and effective method to quickly assess the relative activity and applicability of catalysts. It allows for a rapid understanding of which catalysts exhibit higher catalytic activity at the same temperature. This comparison is particularly valuable for selecting suitable catalysts and optimizing reaction conditions.
Comments 8: Line 267, 268: Why do the reduction peaks of bimetallic catalysts shift to low temperatures?
Response 8: Thanks for the comment. The shift of Pt reduction peaks to lower temperatures in bimetallic PtNi catalysts is attributed to electronic and structural effects induced by the presence of nickel. The incorporation of nickel alters the electronic structure, promotes the formation of a PtNi alloy, and enhances catalytic activity.
Comments 9: Explain the relations used in calculating apparent activation energy.
Response 9: According to the Arrhenius formula: k=Ae-Ea/RT, where k is the rate constant, R is the molar gas constant, T is the thermodynamic temperature, Ea is the apparent activation energy, and A is the prefactor (also known as the frequency factor). This formula can be further expressed as ln k = ln A – Ea/RT, representing the relationship between the reaction rate and reaction temperature. The reaction kinetics were studied with CH4 conversion kept lower than 10% to ensure a kinetically controlled regime and eliminate thermal and diffusion effects.
Minor Comments:
Comments 1: Line 98, 124: inconsistency with Ni precursor notation.
Response 1: Thanks for the comment. We have modified the notation of the Ni precursor in line 124 to Ni(acac)2 in the updated version of line 387 and highlighted the change in red.
Comments 2: Line 198, 201: formatting issue
Response 2: Thanks for the comment. We have further changed the format and fixed some minor errors.
Comments 3: Line 183, 210: incorrect representation of JCPDS file
Response 3: Thanks for the comment. We have incorporated the corrections and highlighted them in red in the revised manuscript. Specifically, Ni (JCPD-04-0850) and Pd (JCPD-46-1043) in line 198 have been updated to Ni (ICDD-JCPDS Card No. 04-0850) and Pd (ICDD-JCPDS Card No. 46-10430), respectively. Additionally, CeO2 (PDF: JCPDS#34-0394) in line 210 has been revised to (ICDD-JCPDS Card No. 34-0394). We appreciate your attention to detail and have ensured that the nomenclature aligns with the International Centre for Diffraction Data (ICDD) standards.
Comments 4: Line 254: Is it HAADF or HADDF? Explain the acronym.
Response 4: Thanks for pointing that out. The correct acronym is HAADF, and we have corrected this error and marked it in red in the revised manuscript.
Comments 5: Line 141: Correct the formula consistent to Figure 7. Conversion should be defined on molar flow basis. Based on chromatographs, the given formula is correct but should be defined scientifically. Give the equation numbering.
Response 5: Thanks for the suggestion and for bringing this to our attention. We have redefined the formula and adjusted the equation number accordingly.
Round 2
Reviewer 2 Report
Comments and Suggestions for Authors
Authors have addressed most of the concerns raised in the previous review report. However, some are not appropriately resolved.
1. Hydrophobic nature is highlighted hypothetically without any proof. Authors can present the experimental proof or remove the discussion around it.
The author's response to comment-1 discusses the PdNi dispersion during the synthesis. After that calcination step was also implemented. So, there is no guarantee that the material still holds hydrophobicity. Thus, any addition of experimental proof helps to confirm the hypothesis.
2. A comparison table on activation energy or TOF, feed concentration, GHSV and T50 must be added.
3. % Conversion formula is not corrected. It should be defined on molar basis. The notations used in the formula should be explained.
Author Response
Response to Reviewer Comments
Thank you for your thorough review of the manuscript once again. We have made detailed revisions addressing the comments, and we believe that this revised version meets the standards for publication in Catalysts.
Point-by-point response to Comments and Suggestions for Authors
Comments 1: Hydrophobic nature is highlighted hypothetically without any proof. Authors can present the experimental proof or remove the discussion around it. The author's response to comment-1 discusses the PdNi dispersion during the synthesis. After that calcination step was also implemented. So, there is no guarantee that the material still holds hydrophobicity. Thus, any addition of experimental proof helps to confirm the hypothesis.
Response 1: Thank you for the valuable suggestions. The cerium dioxide carrier in the prepared catalysts is not hydrophobic; its hydrophobic function is solely utilized for stabilizing the PdNi alloy particles during the preparation process. It can be considered that there is no correlation between hydrophobicity and catalytic performance. As a result, we have revised the title to "Uniformly Dispersed Nano Pd-Ni Oxide Supported on Polyporous CeO2 and Its Application in Methane Conversion of Tail Gas from Dual-fuel Engine," without emphasizing the hydrophobicity of cerium dioxide. Additionally, considering the lack of robust evidence regarding the hydrophobicity of CeO2, we have removed the relevant description.
Comments 2: A comparison table on activation energy or TOF, feed concentration, GHSV, and T50 must be added.
Response 2: Thank you for your valuable comment. We conducted a detailed comparison of Pd loading, feed concentration, gas hourly space velocity (GHSV), and T50. However, we did not include the activation energy or turnover frequency (TOF) due to the lack of description in the selected reported works. The results of this comparison are presented in Table 3 below. From the table, it is evident that the PdNi-H catalyst reported in this manuscript exhibits outstanding superiority in terms of Pd loading and catalytic performance. We have added Table 3 to the manuscript along with a corresponding description in response to your feedback.
Table 3 The comparison of Pd loading, Feed concentration, GHSV, and T50 for reported and this work
Catalysts |
Pd wt.% |
Feed concentration |
GHSV (h-1) |
T50 (℃) |
Reference |
PdCo/Al2O3 |
0.5 |
0.4%CH4/10%O2/N2 |
300000 |
500 |
[9] |
PdNi/Al2O3 |
0.029 |
4100 ppm methane, 5 mol % water |
3128 |
400 |
[15] |
PtPd/ MnLaAl11O19 |
0.67 |
5%CH4 in air |
10000 |
355 |
[41] |
PtPd/Al2O3 |
0.5 |
0.5%CH4/2.0%O2/ Ar |
200000 |
450 |
[42] |
PdNi@Hal |
1 |
1%CH4/20%O2/Ar |
72000 |
355 |
[43] |
Pd/6CoDEG/OMA |
0.5 |
1%CH4/10%O2/N2 |
30000 |
385 |
[44] |
PdNi/H |
0.5 |
1%CH4/5%O2/N2 |
30000 |
351 |
This work |
Comments 3: % Conversion formula is not corrected. It should be defined on molar basis. The notations used in the formula should be explained.
Response 3: Thank you for pointing that out. The CH4 conversion formula has been corrected as follows:
XCH4 = (CCH4(in) −CCH4(out))/ CCH4(in) × 100%
Here, XCH4 represents the CH4 conversion, CCH4(in) denotes the concentration of methane in the feed, and CCH4(out) refers to the concentration of methane at the reactor outlet.